# A Method for Analyzing the Impact of Intra-System and Inter-System Interference on DME Based on Queueing Theory

**DOI:** 10.3390/s19020348

**Published:** 2019-01-16

**Authors:** Guofeng Jiang, Yangyu Fan

**Affiliations:** 1School of Electronics and Information, Northwestern Polytechnical University, Xi’an 710072, China; fan_yangyu@mail.nwpu.edu.cn; 2Aviation Maintenance School for NCO, Air Force Engineering University, Xinyang 464000, China

**Keywords:** DME, JTIDS, intra-system interference, inter-system interference, queueing theory, reply efficiency

## Abstract

In order to use Distance Measuring Equipment (DME) properly, the impact of intra-system and inter-system electromagnetic interference must be analyzed firstly. However, the error of interference analysis using present methods based on pulse overlap is large when there are more aircraft. The aim of this article is to study a method of analyzing interference on DME whether the number of aircraft is small or not. According to the flow chart of DME signal, we studied the limitations of present methods; then constructed a model of analyzing the collision between duration of desired signal and dead time of receiver based on M/M/1/0 queueing system. Combing this model with other methods, we present a analytic model of analyzing intra-system and inter-system interference on DME. Using this analytic model, we analyzed reply efficiency (RE) and capacity of DME under intra-system and Joint Tactical Information Distribution System (JTIDS) interference. The result shows that the calculation for the probability of overlap between DME dead time and subsequent signals using queueing model agrees well with simulation. Consequently, the analytic model is more accurate than using a single method to analyze interference on DME.

## 1. Introduction

In recent years, electromagnetic environment has been becoming more and more complicated. It is very important to analyze electromagnetic compatibility (EMC) between two or more radio equipments operating in the same frequency bands. DME is an aeronautical radio navigation system and has been used for many years [1]. Several aeronautical equipment operate in the same frequency band as DME; for example, Air Traffic Control Radar Beacon System (ATCRBS), Airborne Collision Avoidance System (ACSS), Identification Friend or Foe (IFF) and JTIDS [2]. Many researchers have analyzed EMC between DME and the equipment mentioned above using different methods, for example, Neji et al. [3,4] analyzed impact of L-band Digital Aeronautical Communication System 2 (L-DACS2) on DME under the co-site scenario. Houdzoumis [5] explained the mechanisms of JTIDS interference to DME on first principles. Wolff et al. [6] utilized Systems Tools Kit simulations to gain insight into the power received by a radio occultation satellite in low Earth orbit from a DME transmission as well as the amount of interfering stations. Baek et al. [7] analyzed the pulse collision probability under the worst case by constructing the mathematical models rather than using simulation. Tsinos et al. [8] proposed a novel carrier aggregation scheme to handle the coexistence matters within the unlicensed bands with an efficient decentralized way. Generally speaking, the method of analyzing interference can be classified into two types, one is called the overlap method and the other is signal processing simulation. The power of interfering signal is assumed to be strong enough to interfere with the desired signal when overlap method is used; the interference metric originates in the overlap between desired and interfering signals in the time or frequency domain, and the overlap probability is usually regarded as interference probability. The capability of signal processing of the equipment is ignored, so the result based on overlap method is conservative, but it is the fundamental of interference judgment. Signal processing simulation is based on the principle of equipment and the interfering signal is assumed to be noise, then signal-to-noise ratio (SNR) or bit error rate (BER) is the metric of interference [9]. Using signal processing simulation, interference can be analyzed in the time and frequency and power domains at the same time, so the result is authentic if the flowchart of signal processing and parameters are in accordance with the actual equipment. However, the signal processing and detailed parameters are secret especially for military equipment, and they have to be substituted by typical value; inevitably, there is a deviation from the actual interference. On the contrary, there are only a few parameters (e.g., duty cycle and repetition frequency) used for interference analysis using overlap method. Moreover, these parameters can be obtained easily.

Since the fundamental reason of interference is that interfering signals overlap with the desired signal, overlap method is applied widely to analyze interference in the time domain; overlap is judged by the relative position of rising edge between desired and interfering signal. Analysis of overlap can be achieved by Monte Carlo simulation or mathematical derivation. Using Monte Carlo method, the initial time difference between desired and interfering signal varies randomly in every simulation. Then the average overlap probability is achieved after many simulations. Lo et al. [10] derived the formula of overlap probability between two or more signals, the statistical analytic model of interference analysis is derived from pulse duration and Pulse Repetition Frequency (PRF). Houdzoumis [5] derived the formula of overlap probability based on Poisson distribution of pulse stream composed of desired signal and interfering signals. The error of the models mentioned above is little if the number of signals is not too many and overlap probability is not too big, otherwise, the error is large. The reason is that dead time of receiver is thought to be a part of pulse duration for the entire desired signals. In fact, dead time can be regarded as a part of pulse duration only when the desired signal does not overlap with others. It is assumed that desired signal can be received only when it does not overlap with its adjacent pulses and its arrival time is out of dead time. However, the formulas mentioned above cannot calculate the overlap between arrival and dead time rightly, and consequently the error occurs. Dead time is generated by the receiving of front desired signals, but it affects the receiving of subsequent signals since they are thought to be interfered if their arrival times overlap with dead time; however, it is too difficult to judge easily which one of the subsequent signals is in the dead time. Owing to the similarity between the receiving of signal and queueing process, queueing theory is often applied to analyze the characteristics of communication [11,12,13]. This paper developed a mathematical model for calculating collision probability between desired signal and intra-system interference based on M/M/1/0 queueing theory model, thus, the collision probability between dead time and subsequent pulse signals is calculated accurately even if the pulse-stream density is high, and consequently intra-system interference can be analyzed accurately. Combining the mathematical model of collision probability for periodic pulse, we built a mathematical model to calculate RE for DME when interfered by intra-system and inter-system signals.

The remainder of the paper is organized as follows. In Section 2, we revisited the principles of DME and JTIDS briefly, and focused on the technical characteristics of the two systems that are pertinent to the analysis of interference such as signal structure, pulse duration and PRF. In Section 3, we constructed the flow chart of DME signal firstly, and then defined the evaluation criteria of DME capability. Finally, we built different models to analyze the impact of intra-system and inter-system interference on DME using periodic pulse collision method (PPCM) and theory, respectively. In Section 4, we compared the result from PPCM and theory with that from Monte Carlo simulation, then analyzed RE with respect to capacity, studied capacity of DME when RE and reply rate are limited. In Section 5, we provide conclusions.

## 2. Overview of DME and JTIDS

### 2.1. First Principles and Baseband Signal of DME

DME is an aeronautical radio navigation system and consists of transponder installed on the ground and interrogator fitted in the aircraft. DME can provide the slant range *d* between aircraft and ground beacon continuously. Interrogator transmits interrogations to transponder. After receiving interrogation, transponder delays it for quite a period named as dead time $t_{d}$, transponder refuses to receive other interrogations in the dead time. After the delay of a dead time, interrogations are renamed as reply and transmitted back to interrogator, it takes a period $\tau_{r}$ for interrogator to transmit interrogations and receive reply [1]. The slant range *d* can be calculated as follows:(1)$$d=\frac{\left(\tau_{r}-t_{d}\right)\cdot c}{2}$$
where *c* is velocity of light.

DME signal is a pair of pulses illustrated in Figure 1, the waveform is depicted to be rectangular for simplicity. DME pulse pair characteristics are shown in Table 1. Pulse duration $w_{m}$ is 3.5 μs to all signals, while pulse interval $s_{m}$ is determined by DME mode (X or Y) and signal type (interrogation or reply), which equals to 12 μs or 30 μs or 36 μs. Signal duration $\tau_{d}$ equals to $w_{m}$ plus $s_{m}$. The average PRF of interrogations is (10–30) Hz or (40–150) Hz when interrogator is operating on tracking or searching condition, respectively. The transponder operates at a transmission rate, including randomly distributed pulse pairs and identification signal and distance reply pulse pairs, of no less than 700 pulse pairs per second (ppps) and no more than 2700 ppps.

### 2.2. First Principles and Baseband Signal of JTIDS

JTIDS refers to the communications component of Link 16, which is the designation of a tactical data link that is being fully integrated into operations of the Joint Services, the forces of the North Atlantic Treaty Organization (NATO), and other Allies. JTIDS has the integrated capability of communication, navigation and identification. JTIDS uses the principle of Time Division Multiple Access (TDMA), all JTIDS units are preassigned sets of time slots in which to transmit their data and in which to receive data from other units. Each time slot is 1/128 s, or 7.8125 milliseconds (ms), in duration. The time distribution of a JTIDS slot is shown in Figure 2, there is a random jittering time $t_{J}$ at the beginning of a slot time, then pulse signals time $\tau_{J}$ follows, safeguard time $t_{g}$ is in the last. The single pulse symbol packet consists of one 6.4 μs pulse of modulated carrier followed a 6.6 μs dead time for a total pulse symbol packet duration of 13 μs. The period of pulse signals is composed of 258 pulses when the data pulse format of the time slot is packed in standard double pulse or packed-2 single pulse, thus, the pulse signals lasted 3.354 ms. If the data pulse format of the time slot is packed in packed-2 double pulse or packed-4 single pulse, the period of pulse signals is composed of 444 pulses, jittering time is 0, safeguard time is 2.0405 ms, and then the pulse signal is 5.772 ms [14,15].

### 2.3. Frequency Distribution of DME and JTIDS

The portion of the frequency spectrum between 950 MHz and 1150 MHz is called the Lx band. DME and JTIDS are all operating within the Lx band; the distribution of frequency is shown in Figure 3. DME operates in (962–1213) MHz every 1 MHz, and works in two modes named as X and Y, respectively; there are 126 channels in each mode. The carrier frequency of interrogation in all modes and reply in mode Y is (1025–1150) MHz. The carrier frequency of reply are (962–1024) MHz and (1151–1213) MHz when DME operate in mode X and Y, respectively. So the difference of carrier frequency between interrogation and reply is always 30 MHz.

JTIDS operates mainly in the Lx band between 960 MHz and 1215 MHz. The 51 frequencies assigned to JTIDS for TDMA transmissions are those between 969 MHz and 1206 MHz, 3 MHz apart. The frequency is not held constant during the time slot but is changed rapidly (every 13 microsecond) according to a predetermined pseudorandom pattern; this technique is called frequency hopping.

## 3. Analysis Method of iNterference on DME

### 3.1. Metric of Interference on DME

A DME signal flowchart is shown in Figure 4 during the process of the whole work. There are *n* (*n* > 1) aircraft installed DME interrogator transmit interrogations to the ground transponder at the same time; moreover, JTIDS is in the same electromagnetic environment as DME. All interrogations transmitted from different DME interrogators form an interrogation pulse-stream with pulse density $\lambda_{i}$. Since the initial time and PRF of interrogations are all different, it is unavoidably for pulses in the interrogation pulse-stream to collide with each other. Thus intra-system interference occurs. JTIDS signal is called inter-system interference as far as DME is concerned because they belong to different systems. The density of valid request pulse changes from $\lambda_{i}$ to $\lambda_{i2t}$ after interfered by intra-system interference and JTIDS during the transmission from interrogator to transponder. A period named as dead time immediately following the decoding of a valid interrogation during which a received interrogation will not cause a reply to be generated, dead time is intended to prevent the transponder from replying to echoes resulting from multi-path effects. The subsequent pulse will not be received during dead time, the collision between subsequent pulse and dead time is a part of intra-system interference, then the density of valid request pulse changes from $\lambda_{i2t}$ to $\lambda_{r}$ after overlapping with dead time.

Transponder transmits identification pulse pairs every 40 s at the speed of 1350 ppps besides transmitting request pulses, the maximum period of identification pulse transmitting is 5 s, so the average valid density of identification pulse $\lambda_{id}=\frac{5}{40}\times 1350=169$ ppps. Moreover, identification pulse has priority over request pulse to transmit, that is to say, if the time of identification pulse overlaps with request pulse, then request pulse is thought to be interfered and cannot be received by interrogator, so identification pulse is inter-system interference source too. Reply consisting of identification and request pulses not interfered whose valid density is $\lambda_{t}$ are transmitted from transponder to interrogators, which can be interfered by JTIDS again, the density of valid request pulse is $\lambda_{t2i}$ when they arrive at interrogators. The density of valid request pulse change from $\lambda_{i}$ to $\lambda_{t2i}$ after a whole working process, so the ratio of replies received to the total of transmitted interrogations by the interrogators named as reply efficiency $R_{e}$ can be thought to be indicator for capability of DME. $R_{e}$ can be calculated as follows:(2)$$R_{e}=\frac{\lambda_{t2i}}{\lambda_{i}}=\frac{\lambda_{r}}{\lambda_{i}}\cdot \frac{\lambda_{t}}{\lambda_{r}}\cdot \frac{\lambda_{t2i}}{\lambda_{t}}=P\left(I\right)\cdot P\left(D\right)\cdot P\left(R\right)$$
where *I* denotes the event that valid interrogations are received by transponder, *D* is the event that valid interrogations are not interfered by identification pulses, and *R* is the event that reply transmitted from transponder which is not interfered by JTIDS. The probabilities of these events are P(I), P(D) and P(R), respectively.

### 3.2. Calculation for P(I)

As mentioned above, P(I) is the probability that the interrogation pulses from interrogator output ($\lambda_{i}$) to transponder ($\lambda_{r}$) are not interfered. In this process, intra-system interference consists of collision between two or more interrogations and collision between dead time and subsequent interrogations, inter-system interference is JTIDS. Let *M* denotes the event that intra-system interference does not occur and P(M) is the probability of *M*, *J* denotes the event that JTIDS interference does not occur and P(J) is the probability of *J*. Thus P(I) is given as follows:(3)$$P\left(I\right)=P\left(M,J\right)=P\left(J\right)\cdot P\left(M\left|J\right.\right)$$
where P(M|J) is the conditional probability.

#### 3.2.1. Calculation for P(J)

Figure 3 shows that JTIDS and DME operate within the same band. We assume that only 3 from 51 JTIDS carrier frequencies are deemed potentially harmful to the operation of a DME interrogator, from the total population of potentially harmful JTIDS pulses, we assume in the following that a percentage 60% have sufficient strength to corrupt an overlapping DME pulse [5]. According to [10], P(J) can be calculated as follows:(4)$$P\left(J\right)=1-\frac{3}{51}\cdot 60\%\cdot \frac{\tau_{J}+\tau_{m\_i}}{T_{J}}=1-\frac{3}{85}\cdot \frac{\tau_{J}+\tau_{m\_i}}{T_{J}}$$
where $\tau_{J}$ is duration of pulse symbols defined in Figure 2, $T_{J}$ is repetition period of JTIDS slot equaling to 1/128 s, $\tau_{m_{-}i}$ is DME interrogation pulse pair duration defined as $\tau_{m}$ shown in Figure 2.

#### 3.2.2. Calculation for P(M|J)

When there are *n* aircraft installed DME, 95% of the aircraft are operating in track mode and the others in search mode [10]. So the number of DME in track mode is $L_{t}=\left\lfloor 95\%\cdot n\right\rfloor$, the number of DME in search mode is $L_{s}=\left\lfloor 5\%\cdot n\right\rfloor$, where $\left\lfloor x\right\rfloor$ is the floor function operated on *x*. So interrogation pulse density $\lambda_{i}$ can be calculated as follows:(5)$$\lambda_{i}=L_{t}\cdot f_{t}+L_{s}\cdot f_{s}$$
where $f_{s}$ is PRF of interrogation operating in search mode and $f_{t}$ is PRF of interrogation operating in track mode.

When *n* is small: Since PRF and duty cycle of DME interrogation are all small, the collision probability of interrogations is small accordingly. It can be assumed that dead time is a part of pulse duration, thus, the desired pulse duration equals $\tau_{m}$ plus $t_{d}$. All the interrogations are assumed to be independently at the same time, according to [10], the probability of DME interrogations not interfered $P_{s}$ in search mode is given as follows:
(6)$$P_{s}={\left[1-f_{s}\cdot \left(2\tau_{m\_i}+t_{d}\right)\right]}^{L_{s}-1}\cdot {\left[1-f_{t}\cdot \left(2\tau_{m\_i}+t_{d}\right)\right]}^{L_{t}}$$Similarly, the probability of DME interrogations not interfered $P_{t}$ in track mode is given as follows:
(7)$$P_{t}={\left[1-f_{t}\cdot \left(2\tau_{m\_i}+t_{d}\right)\right]}^{L_{t}-1}\cdot {\left[1-f_{s}\cdot \left(2\tau_{m\_i}+t_{d}\right)\right]}^{L_{s}}$$Combining Equations (5)–(7), the average probability of multi-path DME interrogations not interfered P(M) can be obtained as follows:
(8)$$P\left(M\right)=\frac{P_{t}\cdot L_{t}\cdot f_{t}+P_{s}\cdot L_{s}\cdot f_{s}}{\lambda_{i}}$$Event *M* and event *J* are assumed to be statistically independent, that is to say, inter-system and intra-system interference are independent of each other. Substituting Equations (4) and (8) into (3), we obtain
(9)$$P\left(I\right)=P\left(J\right)\cdot P\left(M\left|J\right.\right)=P\left(J\right)\cdot P\left(M\right)=\left(1-\frac{3}{85}\cdot \frac{\tau_{J}+\tau_{m\_i}}{T_{J}}\right)\cdot {\left[1-\lambda_{i}\cdot \left(2\tau_{m\_i}+t_{d}\right)\right]}^{n-1}$$The collision probability between two or more DME interrogations will increase when *n* is large. The result based on Equation (Equation 7) is smaller than actual value. Moreover, intra-system interference is correlative with inter-system interference, the error using Equation (Equation 9) is large if *n* is large. Equation (Equation 9) can be apply to a small quantity of aircraft only.When *n* is large: We can adopt simplifying assumption that these interrogations are randomly distributed with respect to time forming a Poisson process [5], considering that interrogations will be abandoned if its arrival time overlaps with dead time generated by front interrogation, and hence the receiving process of DME interrogations can be regarded as a quasi M/D/1/0 theory model. In such a birth-and-death model M/D/1/0, the interrogation pulses are customers, transponder is service facility, “M” represents that the arrivals occur from an infinite source in accordance with a Poisson process with parameter $\lambda_{i2t}$ defined in Figure 4—that is, the inter-arrival times are independent exponential with mean 1/$\lambda_{i2t}$, “D” represents service times that are deterministic and equivalent to dead time $t_{d}$ plus $\tau_{m_{-}i}$, “1” represents single server, and “0” means that customer will depart when its arrival time overlaps with service time [16]. Service rate μ is given as follows:
(10)$$\mu =\frac{1}{\tau_{m\_i}+t_{d}}$$According to [11], the probability of customers serviced *Q* can be calculated as follows:
(11)$$Q=\frac{\mu }{\lambda_{i2t}+\mu }=\frac{1}{\lambda_{i2t}\cdot \left(\tau_{m\_i}+t_{d}\right)+1}$$
where the density of interrogation pulse-stream $\lambda_{i2t}=P\left(J\right)\cdot \lambda_{i}$ is subject to $P\left(J\right)$, *Q* is condition probability. The subsequent customer cannot affect the front service in a normal M/D/1/0 theory model; however, if the subsequent interrogation pulse overlaps with the front interrogation pulse before it enters into a transponder, the front interrogation pulse will be abandoned too, which is different from a normal birth-and-death model. Considering the interrogation duration on its back side, the probability of an interrogation not overlapped with its subsequent interrogations is calculated to be $e^{-\lambda_{i2t}\cdot \tau_{m\_i}}$. With the help of queueing theory model, P(M|J) can be calculated as follows:
(12)$$P\left(M\left|J\right.\right)=e^{-\lambda_{i2t}\cdot \tau_{m\_i}}\cdot Q=\frac{e^{-\lambda_{i2t}\cdot \tau_{m\_i}}}{\lambda_{i2t}\cdot \left(\tau_{m\_i}+t_{d}\right)+1}$$Substituting Equations (4) and (12) into (3) gives:
(13)$$P\left(I\right)=\frac{e^{-\lambda_{i}\cdot \left(1-\frac{3}{85}\cdot \frac{\tau_{J}+\tau_{m\_i}}{T_{J}}\right)\cdot \tau_{m\_i}}\cdot \left(1-\frac{3}{85}\cdot \frac{\tau_{J}+\tau_{m\_i}}{T_{J}}\right)}{\lambda_{i}\cdot \left(1-\frac{3}{85}\cdot \frac{\tau_{J}+\tau_{m\_i}}{T_{J}}\right)\cdot \left(t_{d}+\tau_{m\_i}\right)+1}$$

### 3.3. Calculation for P(D)

As mentioned above, reply pulses consist of measuring and identification pulses, moreover, identification pulses has priority to transmit. What we are concerned with is whether measuring pulses is interfered, and identification pulses can be thought to be inter-system interference. Interference probability can be calculated as [10], since the characteristic of identification pulses is the same as that of measuring pulses, the probability of measuring pulses not interfered by identification pulses $P\left(D\right)$ is given as follows:(14)$$P\left(D\right)=1-\lambda_{id}\cdot \left(\tau_{m\_r}+\tau_{m\_r}\right)=1-\frac{5}{40}\times 1350\times 2\tau_{m\_r}=1-337.5\cdot \tau_{m\_r}$$
where $\tau_{m_{-}r}$ is the reply pulse pair duration defined as $\tau_{m}$ shown in Figure 1, its value can be seen from Table 1.

### 3.4. Calculation for P(R)

After DME transponder receives a valid interrogation, delays it a dead time, then transmit back to interrogator; meanwhile, PRF of DME interrogations is jittered to ensure that interrogator can recognize their own reply pulses only, thus, there is no intra-system interference in DME reply pulses. They can be interfered by inter-system interference sources (e.g., JTIDS) only just shown in Figure 4, according to the probability formula for inter-system interference, P(R) can be calculated as follows:(15)$$\;P\left(R\right)=1-\frac{3}{85}\cdot \frac{\tau_{J}+\tau_{m\_r}}{T_{J}}$$
when the number of aircraft *n* is small, if we make substitution of Equations (9), (14) and (15) into (2), then we have:(16)$$\;R_{e}=\left(1-\frac{3}{85}\cdot \frac{\tau_{J}+\tau_{m\_i}}{T_{J}}\right)\cdot \frac{P_{t}\cdot L_{t}\cdot f_{t}+P_{s}\cdot L_{s}\cdot f_{s}}{L_{t}\cdot f_{t}+L_{s}\cdot f_{s}}\cdot \left(1-337.5\cdot \tau_{m\_i}\right)\cdot \left(1-\frac{3}{85}\cdot \frac{\tau_{J}+\tau_{m\_r}}{T_{J}}\right)$$
when the number of aircraft *n* is large, substituting Equations (13)–(15) into (2), it follows that
(17)$$R_{e}=\frac{e^{-\lambda_{i}\cdot \left(1-\frac{3}{85}\cdot \frac{\tau_{J}+\tau_{m\_i}}{T_{J}}\right)\cdot \tau_{m\_i}}\cdot \left(1-\frac{3}{85}\cdot \frac{\tau_{J}+\tau_{m\_i}}{T_{J}}\right)\cdot \left(1-337.5\cdot \tau_{m\_r}\right)\cdot \left(1-\frac{3}{85}\cdot \frac{\tau_{J}+\tau_{m\_r}}{T_{J}}\right)}{\lambda_{i}\cdot \left(1-\frac{3}{85}\cdot \frac{\tau_{J}+\tau_{m\_i}}{T_{J}}\right)\cdot \left(t_{d}+\tau_{m\_i}\right)+1}$$

## 4. Results and Discussion

### 4.1. The Result of Re Based on Different Methods

To verify the accuracy of the analytical model formulated as Equation (Equation 17) presented in this paper, regardless of the limitations of transponder reply rate and DME capacity, we simulate and calculate *Re* based on Equations (16), (17) and a Monte Carlo method. For simplicity, we assume that DME signals are thought to be valid unless they are collided. Simultaneously, DME signals can be received and processed as long as they are valid signals. Simulation parameters are shown as follows:$f_{t}\;=15\;\mathrm{Hz}$, $f_{s}\;=150\;\mathrm{Hz}$.95% of aircraft are in track mode and others are in search mode.DME operates in mode X, $\tau_{m\_i}\;=\;\tau_{m\_r}=15.5\;\mu$s and $t_{d}\;=60\;\mu$s.PRF of identification pulse: $f_{id}\;=169\;\mathrm{Hz}$.Inter-system interference is absent.Simulation time: 1 s.Number of aircraft: variation from 5 to 600 every 3.Number of Monte Carlo simulation: 1000.

Figure 5 plots the analytic and simulation results of RE (equivalent to probability of reply) versus the number of aircraft. The numbers on the curve are interrogation pulse density $\lambda_{i}$ as defined in Figure 4, for example, the first number 75 means $\lambda_{i}=75$ pulses per second. Figure 5 shows that $\lambda_{i}$ increases with the number of aircraft; accordingly, DME intra-system interference increases with $\lambda_{i}$; as a result, RE decreases with a rise in the number of aircraft just as shown in Figure 5. It can be seen that the results based on Equation (Equation 16) are consistent with simulations only when the number of aircraft is less than about 100, the difference of results based on Equation (Equation 16) and simulations increases with the number of aircraft, which results from the assumption in the derivation of Equation (Equation 16). As noted in the previous section, Equation (Equation 16) assumes that none of the signals overlap with dead time. However, collision probability increases with the number of aircraft, such assumptions are not supported at all; therefore the error rises with the increase of number of aircraft.

Figure 5 shows that the difference between results based on Equation (Equation 17) and simulations increases with the number of aircraft, but the results based on Equation (Equation 17) are only 0.6% and 2.1% less than simulation when the number of aircraft are 100 and 575, respectively; however, the result based on Equation (Equation 16) is 12.3% less than simulation when the number of aircraft is 575. In general, the results based on Equation (Equation 17) agree well with simulation; the reason is that queueing theory model solves the problem of intra-system interference compared to Equation (Equation 16); hence, the error of Equation (Equation 17) is less than that of Equation (Equation 16) when the number of aircraft is large. Note, Equation (Equation 17) is based on a large number of aircraft; in fact, owning to the small duty cycle and PRF of DME interrogation, when the number of aircraft is small, intra-system interference is very little; in conclusion, Equation (Equation 17) can be used whether the number of aircraft is small or not.

### 4.2. Analysis of JTIDS Interference and DME Mode on RE

DME is interfered by inter-system and intra-system interference at the same time, as has been discussed above, the probability of inter-system interference are P(D) and P(R), and P(R) is the probability of intra-system interference. Some simulation parameters are the same as mentioned above, others are shown as follows:When DME operates in mode Y, $\tau_{m\_i}\;=39.5\;\mu$s, $\tau_{m\_r}\;=33.5\;\mu$s and $t_{d}\;=66\;\mu$s.JTIDS signal duration: $\tau_{J}\;=\;5.772$ ms.Number of aircraft: variation from 5 to 400 every 15.$f_{J}=0$ means that there is no JTIDS interference, $f_{J}=128$ means that JTIDS is in present.

When the number of aircraft varies from 5 to 400 with 15 spacing, DME RE based on Equation (Equation 17) is illustrated in Figure 6, where mode X and Y indicate that DME operates in mode X and Y, respectively. The numbers on the curve marks are DME transponder reply rate accordingly.

Figure 6 shows that DME reply efficiency decreases obviously when interfered by JTIDS compared to no JTIDS interference, the decrease value is about 5% when there are 5 aircraft. However, RE decreases smaller as the number of aircraft increases, the decrease value is about 2% when the number of aircraft is 395. The reason is that inter-system interference is basically unchanged when the number of aircraft changes, but intra-system interference increases with the number of aircraft. Intra-system interference is small when there are small numbers of aircraft, so inter-system interference is the main interference. But as the number of aircraft increases, the main interference is converted from inter-system interference to intra-system interference. As a result, the difference of RE between JTIDS in present and JTIDS in absent becomes smaller and smaller as the number of aircraft increases.

Since the calculation for reply efficiency is based on pulse collision, and DME signal duration in mode Y is larger than that in mode X from Table 1, the collision probability in mode Y is larger than that in mode X according to Equations (13)–(15); as a result, reply efficiency in mode Y is smaller than that in mode X just as shown in Figure 6. Moreover, we can see that the difference of reply efficiency between mode X and Y increases with the number of aircraft, and yet the differences is not always increases in fact. According to Equation (Equation 17), the difference $\Delta R_{e}$ can be calculated as follows:(18)$$\Delta R_{e}=R_{eX}-R_{eY}=\frac{e^{-\lambda_{i}\;\times \;1.5095\;\times \;10^{-5}}\times 0.9434}{\lambda_{i}\times 7.3526\times 10^{-5}+1}-\frac{e^{-\lambda_{i}\;\times \;3.8463\;\times \;10^{-5}}\times 0.9404}{\lambda_{i}\times 1.0273\times 10^{-4}+1}$$
where $R_{eX}$ and $R_{eY}$ are reply efficiencies in mode X and Y, respectively. If the limitations of pulse density $\lambda_{i}$ are not considered, $\Delta R_{e}$ with respect to the number of aircraft is shown in Figure 7. It can be seen that if the number of aircraft is less than about 500, $\Delta R_{e}$ increases with the number of aircraft, and yet if the number is more than 500, $\Delta R_{e}$ decreases. In the present practice, DME capacity is usually less than 200 owning to the limitations of reply efficiency and reply rate.

### 4.3. Impact of Reply Rate and RE on DME Capacity

International civil aviation organization (ICAO) recommends that reply efficiency should be more than 70% and reply rate no less than 2700 ppps to ensure accuracy of distant measuring [1]. Transponder can transmit distant measuring pulses with $\lambda_{t}=2700-169=2531$ ppps pulse density only besides identification pulses as mentioned above. Under the limitations, DME capacity and the corresponding parameters can be obtained from Figure 6 and some of them are listed in Table 2 to compare.

Under the limitation of 2700 ppps reply rate, in mode X, DME capacities are 159 and 154 when there is JTIDS interference or not, respectively. Corresponding reply efficiencies are 71.4% and 75.3%; they are more than 70%, so 159 and 154 are the valid capacities of DME in mode X. DME capacities in mode Y are 216 and 210, they look like more than those in mode X; but they are invalid values because their corresponding reply efficiencies are 52.5% and 55.3%, they are less than 70%. Under the limitation of 70% reply efficiency, DME valid capacities in mode Y are 100 and 117 actually, since their corresponding reply rates are 1748 and 1964, respectively; and they are less than 2700. It looks like a strange result that DME capacity in mode X interfered by JTIDS (159) is more than that without interference (154). However, we should not look DME capacity only, simultaneously, we should combine the corresponding RE. RE interfered by JTIDS (75.3%) is more than without interference (71.4%). So we should consider the two limitations of reply efficiency and reply rate at the same time when calculating DME capacity.

## 5. Conclusions

Combining an M/M/1/0 queueing model with PPCM, we construct an analytical model of calculating overlap probability between two or more pulses. No matter how many aircraft there are, the analytical model can be used to analyze intra-system and inter-system interference on DME. According to the results mentioned in Section 4, we may reasonably conclude that the M/M/1/0 queueing model can calculate the collision probability between dead time of transponder and duration of subsequent desired signals accurately. Hence, the analytical model is more consistent with simulation than single PPCM.

With the rise of the number of aircraft, intra-system interference increases, on the contrary, RE decreases. In addition, RE is affected by inter-system interference; for example, JTIDS interference causes RE to decrease about 5% when there are about 100 aircraft. DME capacity is related to reply rate and RE simultaneously, they are interrelated and interact on each other. In accordance with the limits of RE and reply rate recommended by ICAO, the overall performance of DME cannot be improved unless multi parameters are optimized systematically.

## Figures and Tables

**Figure 1 sensors-19-00348-f001:**
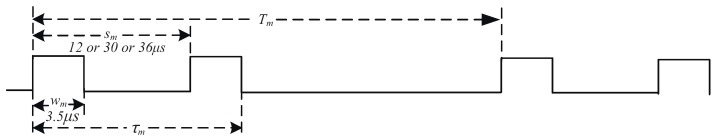
The basic pulse pair of DME signal.

**Figure 2 sensors-19-00348-f002:**
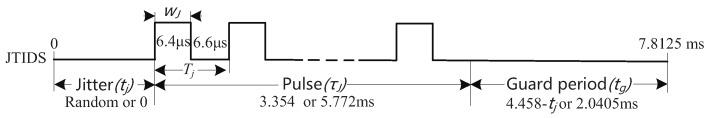
The time distribution of a JTIDS slot.

**Figure 3 sensors-19-00348-f003:**
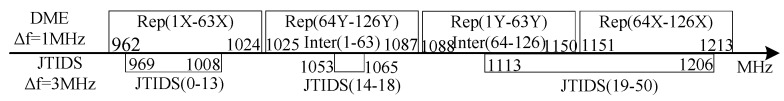
The frequency distribution for DME and JTIDS.

**Figure 4 sensors-19-00348-f004:**
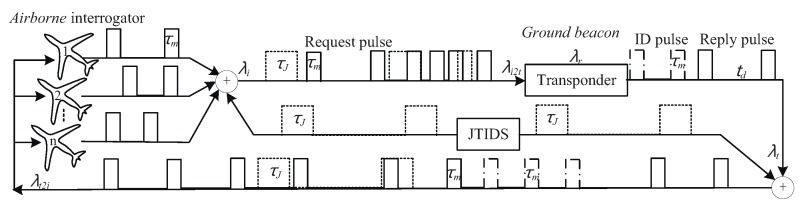
The signal flowchart of DME suffering from inter-system and intra-system interference.

**Figure 5 sensors-19-00348-f005:**
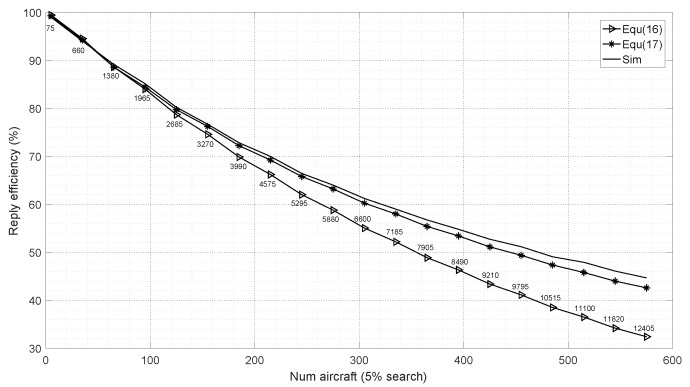
Number of aircraft vs. reply efficiency.

**Figure 6 sensors-19-00348-f006:**
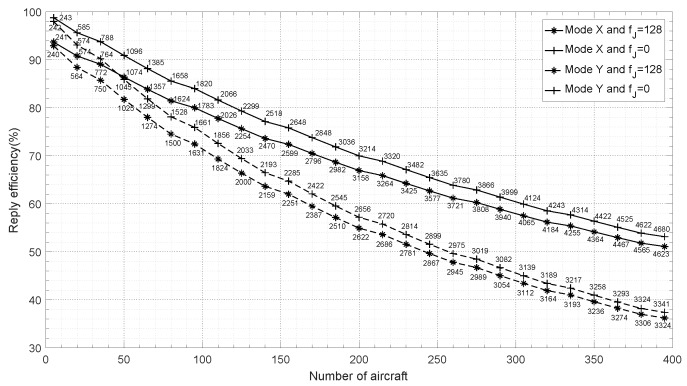
Reply efficiency in different modes vs. number of aircraft.

**Figure 7 sensors-19-00348-f007:**
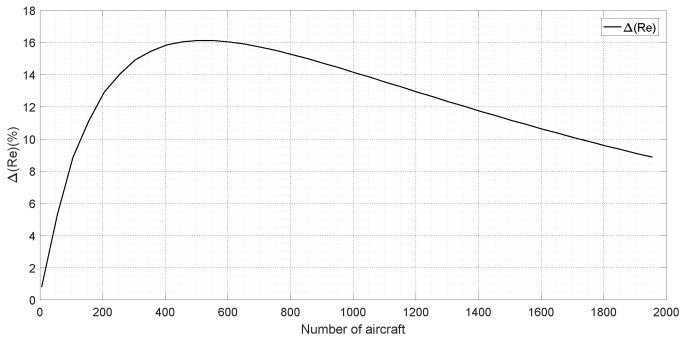
Difference of reply efficiency for different modes vs. number of aircraft.

**Table 1 sensors-19-00348-t001:** DME pulse pair characteristics.

Signal Type	Operation Mode	Pulse Duration $w_{m}$ (μs)	Pulse Interval $s_{m}$ (μs)	Signal Duration $\tau_{m}$ (μs)	PRF(Hz)	
					Tracking	Searching
Interrogation	X	3.5	12	15.5	10–30	40–150
	Y		36	39.5		
Reply	X		12	15.5	700–2700	
	Y		30	33.5		

**Table 2 sensors-19-00348-t002:** DME Capacity under the limitations of reply rate and reply efficiency.

Reply Rate (ppps)	JTIDS	Mode	$R_{e}$	Number of Aircraft
2700	Yes	X	71.4%	***159***
		Y	52.5%	216
	No	X	75.3%	***154***
		Y	55.3%	210
2848	Yes	X	70%	171
1748		Y		***100***
3214	No	X		200
1964	No	Y		***117***

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
