# Peer review of "A Method for Analyzing the Impact of Intra-System and Inter-System Interference on DME Based on Queueing Theory"

_sensors, 2019, doi:10.3390/s19020348_

Reviewer 1 Report

In this work, a method for analyzing the impact of the intra system and JTIDS interference on DME navigation systems is proposed. Analytical results based on the queuing theory are presented and verified by simulations. The contributions of the manuscript are clear and the flow easy to follow. I have the following comments.

1.    The basic problem of the manuscript is that, it is very poorly written. There are lots of grammar and syntax errors and this is totally unacceptable for a manuscript to be published in a journal.

2.    The simulation’s setup has to be explained in more detail, since the provided information is not enough for someone who will want to reproduce the results.

3.    In the beginning of the introductions, the authors make a brief reference to the importance of analyzing the interference of co-existing systems. There, they could also make a short reference to wireless communications systems like cognitive radio, LTE-unlicensed systems which in principle are examining the coexistence of different systems in the same frequency bands. Please find below recommended references

C. G. Tsinos, F. Foukalas and T. A. Tsiftsis, "Resource Allocation for Licensed/Unlicensed Carrier Aggregation MIMO Systems," in IEEE Transactions on Communications, vol. 65, no. 9, pp. 3765-3779, Sept. 2017.

Reviewer 2 Report

From figure 3, there is other interference source in the same band as DME such as ADS-B located in 1090 MHz. Does the method also analyze the non-pulse type of interference?

The reviewer sees the number on the curve in the figure 5. What do the number indicate?

Reviewer 3 Report

General Comments:

The authors propose to explore the EMI between the Distance Measuring Equipment (DME) and the Joint Tactical Information Distribution System (JTIDS) based on M/M/1/0 queuing model, which allow them to play easily include the number of aircrafts as a parameter to evaluate the performance of the DME in terms of reply efficiency. The approach adopted focus on the overlap of signals to evaluate the level of interference between systems. Both intersystem and intrasystem interference is considered on the study which is carried through overlap probability.

Strong Aspects:

·     The main contribution of this paper lies on the formulation of the EMI problem based on queueing theory.

Weak Aspects and Suggestions:

·     The prefixes “intra” and “inter” are wrongly employed. “intra” means between, where as “inter” means within. 

·     Due to the difference of modes provided by the DME system, it would be nice to have on the paper an evaluation of the percentage of aircrafts on track/search mode for a given number of aircrafts, e.g., a result for when 500 aircrafts are considered.

Author Response

Round  2

Reviewer 1 Report

The authors have addressed my comments and thus, I recommend the manuscript acceptance.